# Idiopathic Short Stature: What to Expect from Genomic Investigations

Nathalia Liberatoscioli Menezes Andrade [1], Laurana Polli Cellin [1], Raissa Carneiro Rezende [1], Gabriela Andrade Vasques [2] and Alexander Augusto Lima Jorge [2,*]

[1] Unidade de Endocrinologia Genetica (LIM25), Hospital das Clínicas da Faculdade de Medicina, Universidade de São Paulo (USP), São Paulo 05403-000, Brazil

[2] Unidade de Endocrinologia Genetica (LIM25) e Unidade de Endocrinologia do Desenvolvimento, Laboratorio de Hormonios e Genetica Molecular (LIM42), Hospital das Clinicas da Faculdade de Medicina, Universidade de São Paulo (USP), São Paulo 05403-000, Brazil

* Correspondence: alexj@usp.br

**Abstract:** Short stature is a common concern for physicians caring for children. In traditional investigations, about 70% of children are healthy, without producing clinical and laboratory findings that justify their growth disorder, being classified as having constitutional short stature or idiopathic short stature (ISS). In such scenarios, the genetic approach has emerged as a great potential method to understand ISS. Over the last 30 years, several genes have been identified as being responsible for isolated short stature, with almost all of them being inherited in an autosomal-dominant pattern. Most of these defects are in genes related to the growth plate, followed by genes related to the growth hormone (GH)–insulin-like growth factor 1 (*IGF1*) axis and RAS-MAPK pathway. These patients usually do not have a specific phenotype, which hinders the use of a candidate gene approach. Through multigene sequencing analyses, it has been possible to provide an answer for short stature in 10–30% of these cases, with great impacts on treatment and follow-up, allowing the application of the concept of precision medicine in patients with ISS. This review highlights the historic aspects and provides an update on the monogenic causes of idiopathic short stature and suggests what to expect from genomic investigations in this field.

**Keywords:** short stature; idiopathic short stature; genetic analysis

## 1. Introduction

Short stature is a common concern for physicians caring for children and is defined as having a height exceeding 2 standard deviations (SD) below the corresponding mean for a given age, sex, and population group. Based on this definition, short stature affects 2.3% of children [1]. In the traditional approach taken to investigate the cause of growth deficiencies, these children undergo an extensive medical evaluation searching for hormonal etiologies or other chronic medical conditions. However, previous studies have shown that a standard medical evaluation does not often lead to a diagnosis in a healthy child whose only positive finding based on anamnesis and a physical examination is short stature [2,3]. In about 70% of cases of short stature evaluations, the children are healthy, without clinical and laboratory findings that explain their growth disorder, being classified as having constitutional short stature or idiopathic short stature (ISS) [2,4,5]. This scenario has been changing with the improvement of the genetic sequencing techniques, which have made it possible to diagnose 10–30% of these children, providing an answer to the cause of their isolated short stature [6–13]. This review provides an update on the monogenic causes of idiopathic short stature and indicates what to expect from the genomic investigation in this field.

## 2. Materials and Methods

### 2.1. Search Strategy

We conducted a systematic literature review to identify studies reporting the use of genetic sequencing approaches in the investigation of idiopathic short stature. We searched the PubMed® databases and included studies from 1960 to November 2022.

### 2.2. Eligibility Criteria

We searched the literature for review, observational studies, and clinical studies that included genetic data about children classified as having idiopathic short stature. The key words "idiopathic short stature" and "genetics" were screened, and 6421 results were found. We excluded in this analysis books, documents, and papers that were not written in English, with 1026 results remaining. We considered as the most relevant articles those that encompassed large cohorts and used a multigene sequencing analysis for idiopathic short stature. We chose articles that addressed the term idiopathic short stature for the first time to discuss historical aspects and the most relevant articles that addressed genetic aspects related to idiopathic short stature in the last 20 years.

## 3. Historic Aspects of Idiopathic Short Stature

The term idiopathic is often used to relate to or to denote any disease or condition that arises spontaneously or without any known cause [14]. It is usually a diagnosis of exclusion, but it is unclear what minimum investigation should be carried out to define a condition as idiopathic. Therefore, many medical specialties have difficulties in defining this term [15]. In the context of short stature, it is hard to determine when the term idiopathic was first used in the medical literature concerning children with a growth disorder. To the best of our knowledge, the first time that this concept was reported was in 1964 in a paper that described the growth response to pituitary growth hormone (GH) [16]. In this study, the author explained the following: "The remainder of the patients could presumably be forced into various ill-defined groups, called primordial, ateliotic, constitutional, hypocaloric, or low-birth-weight dwarfs [16]. However, there is still much doubt if these terms delineate what has been posteriorly called "idiopathic growth failure". In 1968, Edelmann et al. used the term "idiopathic growth retardation", referring to children with growth retardation, with no determined etiology [17]. In the early 1970s, the term idiopathic short stature (ISS) appeared in the medical literature for the first time [18,19]. However, the term included patients with short stature regardless of birth weight or length, and it was generally used to describe growth disorders of an unknown cause. In 1996, after a consensus meeting, it was defined that children with ISS would be those with adequate birth weight and length for gestational age (>−2 SDS), without body disproportion, without evidence of chronic organic diseases, without psychiatric illnesses, with adequate nutrition, and without evidence of endocrine deficiencies, including GH [20]. Since then, the term has been widely used in clinical practice and was reinforced by the consensus study published in 2008 [21]. Nevertheless, these consensuses preceded the expansion of genetic studies in growth disorders [22]. Nowadays, more than 900 papers on PubMed address genetic studies involving children with ISS. Several of these studies described genes in which rare deleterious variants cause short stature without other characteristics or only with subtle phenotypes. These genes are mainly involved in the GH/insulin-like growth factor (*IGF-1*) axis and growth plate physiology [23], challenging the continued use of the term ISS.

## 4. Genetic Basis of Idiopathic Short Stature

Human height has a high degree of heritability, indicating that genetic factors are one of the main determinants of height [24]. Environmental aspects also contribute to height variability. Studies that evaluated the growth pattern of twins showed an increasing pattern of height heritability with age, being low in early childhood and up to 0.68–0.94 at older ages [25]. This genetic influence on height is attributed to a complex combination of multiple genes and variants in a typical pattern of a polygenic trait.

Based on these concepts, the genetic evaluation emerged as a great potential tool to understand ISS. In recent years, several genome-wide association studies (GWAS) have identified more than 12,000 common variants affecting adult height [21,24,26]. Each of these common variants (minor allele frequency (MAF) $\geq$ 5%) has a low impact on height variability (less than 1–2 mm), whereas rare variants (MAF < 1%) were associated with having a greater impact on height (around 20 mm) [27,28]. Thus, the combination of common and rare variants explains the height variability in the general population and those with short stature without other clinical manifestations in a polygenic model that is classically used to explain ISS [26,29].

In contrast to the polygenic nature presented above, more extreme phenotypes of short stature or syndromic growth disorders are often associated with monogenic conditions [30,31]. The first described monogenic cause of short stature was severe isolated growth hormone (GH) deficiency. In 1981, it was shown that a deletion of the *GH1* gene caused complete GH deficiency and severe short stature [32]. In the last 40 years, hundreds of genes involved in syndromes associated with growth impairment have been identified [31]. These children and their families carried extremely rare variants with a great effect on the phenotype, usually with syndromic characteristics and Mendelian inheritance [28]. However, over the last 30 years, several genes have been identified as responsible for isolated short stature, almost all of them following an autosomal-dominant inheritance pattern [6,33–47] (Table 1). This background is helping us better understand the genetic basis of short stature in non-syndromic individuals.

**Table 1.** The genes related to "idiopathic short stature" as a monogenic component.

| Gene | Year of Description in ISS | Prevalence in ISS | Evidence | Observation | Ref |
|---|---|---|---|---|---|
| *GHR* | 1995 | 5% | Limited | Laboratory findings suggestive of partial GH insensitivity | [33,48] |
| *SHOX* | 1997 | 2.4% | Definitive | Short stature with mild body disproportion | [49] |
| *GH1* | 2003 | No data | Limited | Severe postnatal short stature, normal to elevated GH peaks in stimulation tests, and low levels of *IGF-1* and IGFBP-3 | [35] |
| *IGF1R* | 2003 | No data | Moderated | The majority were born small for their gestational age and had high levels of *IGF-1* | [36,50,51] |
| *IGFALS* | 2004 | No data | Limited | Severe deficiency of *IGF1* and IGFBP-3, disproportional to the severity of short stature | [52] |
| *GHSR* | 2006 | Only case report | Limited | Associated with GHD and short stature in the same family | [38,53] |
| *NPR2* | 2013 | 2 to 13.6% | Strong | Proportional or disproportional short stature and unspecified skeletal findings at time of X-ray of bone age | [39] |
| *PTPN11* | 2013 | Only case report | Limited | Short stature with or without mild phenotype of Noonan syndrome | [6] |
| *IGF1* | 2013 | No data | Limited | Variable degree of pre- and postnatal growth retardation with serum *IGF-1* at the lower end of normal | [40] |
| *ACAN* | 2014 | No data | Moderated | Short stature with or without advanced bone age | [41] |

| Gene | Year of Description in ISS | Prevalence in ISS | Evidence | Observation | Ref |
|------|---------------------------|-------------------|----------|-------------|-----|
| *FGFR3* | 2015 | Only one case report | Limited | Only one family described with proportional short stature | [42] |
| *PAPP-A2* | 2016 | Only case report | Limited | High levels of *IGF-1* and IGFPB-3 | [43] |
| *NPPC* | 2018 | No data | Limited | Brachydactyly | [44] |
| *IHH* | 2018 | 3.3% | Moderated | Proportional or disproportional short stature, with or without unspecific skeletal findings as shortening of middle phalanges of the 2nd and 5th digits | [45] |
| *STAT5B* | 2018 | Only case report | Limited | Eczema and laboratory results suggestive of partial GH insensitivity | [46] |
| *NF1* | 2019 | No data | Limited | Short stature without clinical features of neurofibromatosis | [54] |
| *COL2A1* | 2021 | 5.7% | Limited | Unspecific skeletal findings and disproportionate short stature | [47] |

## 5. Genes Related to "Idiopathic Short Stature": Monogenic Conditions

### 5.1. Genes Related to the GH-IGF1 Axis

The GH/*IGF-1* axis is an essential postnatal growth regulator. One of the main criteria for classifying a child as having ISS is to exclude a diagnosis of GH deficiency [55]. The other axis defects are rare conditions, often causing extreme short stature and likely to be well characterized by the hormonal profile [56]. However, defects involving this axis can also cause remaining growth disorder phenotypes, often labeled as ISS.

#### 5.1.1. GH1

In addition to the inherent difficulty in ruling out GH deficiency in children with short stature [22,57], the role of bioactive GH molecules has long been postulated [58,59]. Specific mutations in *GH1* make GH unable to activate its receptor but it retains its ability to be detected by GH assays. Few patients with biologically inactive GH have been described in the literature [60–63]. The patients usually have severe postnatal short stature (height SDS < −3) without significant dysmorphisms. A laboratory evaluation of these patients demonstrated a normal to elevated GH peak in stimulation tests and low levels of *IGF-1* and IGFBP-3. The improvement in growth velocity and the increase in *IGF-1* levels during short- and long-term rhGH therapy differentiates these children from those with insensitivity to GH [62]. Despite representing an extreme form of short stature, these patients can be classified as ISS, and only the identification of the genetic variant in *GH1* allows the correct determination of the etiology of the growth disorder [35].

#### 5.1.2. GHR

Classically, biallelic mutations in the GH receptor gene (*GHR*) cause complete insensitivity to GH (GHI) or Laron syndrome, a severe inability to respond to endogenous or exogenous GH with particular growth and metabolic effects [64]. Patients with the classical GHI phenotype have severe postnatal short stature, elevated basal and stimulated GH levels, and extremely low *IGF-1*/IGFBP-3 levels that do not increase after rhGH use. Studies in the 1990s suggested that children classified as having ISS could have partial forms of GHI, with low *IGF-1* and GH binding protein (GHBP) values and high mean 12-h GH values [65]. In humans, GHBP is generated by the cleavage of GHR found on the cell surface, which reflects the expression of this receptor. Over the years, it has become clear that GH insensitivity occurs across a broad spectrum of severity [66]. In 1994, a *GHR* mutation was

first reported in patients classified as ISS [33]. In this study, the author evaluated *GHR* in 14 ISS children with low GHBP and identified three patients with heterozygous and one with compound heterozygous *GHR* variants. Following this study, several others described patients with ISS and heterozygous variants of *GHR*. However, many of these variants would not be classified as pathogenic under the current criteria in variant analyses [67]. As *GHR* haploinsufficiency causes a slight reduction in height (around 4 cm) [68], only variants with a dominant negative effect can relate to short stature with autosomal dominant inheritance [69].

### 5.1.3. *GHSR*

Ghrelin is a brain–gut peptide that acts by binding to the growth hormone secretagogue receptor (*GHSR*) [70]. Beyond its action in energy metabolism and the regulation of appetite, this peptide was also related to bone formation and growth hormone release. The active form of *GHSR* is expressed predominantly in the pituitary and hypothalamus. Studies suggest that ghrelin regulates GH secretion, through *GHSR*, in an endocrine pathway distinct from GHRH and somatostatin, and contributes to the control of pulsatile GH release [71], with a high degree of constitutive activity in vitro. In 2006, Pantel et al. reported a *GHSR* missense variant in two unrelated families with short stature, demonstrating that this defect resulted in decreased receptor expression and selectively impaired the constitutive activity of the *GHSR*, whereas it preserved the ability for ghrelin response [38]. Homozygous and heterozygous defects in the *GHSR* gene were associated with ISS and with GH deficiencies of variable severity and penetrance. Both phenotypes were even described in the same family [38]. Additionally, heterozygous *GHSR* defects were also related to constitutional delays in growth and puberty [72]. This heterogeneity in the clinical presentation may be partially explained by the gene defects causing protein alterations that affect the binding affinity to ghrelin and the constitutive activity of the receptor itself. However, the role of *GHSR* variants in growth disorders is still a matter of debate.

### 5.1.4. *STAT5B*

The signal transducer and activator of transcription 5B (*STAT5B*) is an important cytoplasmic protein in the *GHR* signaling pathway. *STAT5B* is responsible for increases in the expression of several GHR-stimulated genes, such as *IGF1*, its binding protein (IGFBP-3), and the acid labile subunit (ALS) [73]. A homozygous inactivating mutation in the *STAT5B* gene was first described in 2003 as a rare cause of growth hormone insensitivity in a patient with severe short stature and immunodysfunction [74]. Individuals heterozygous for loss-of-function (LoF) mutations in *STAT5B* have a mean negative impact of 3.9 cm in their height, usually resulting in their stature being within the normal range [75]. Collaborating with the idea of a spectrum of effects on height, in 2018, three heterozygous *STAT5B* mutations with dominant negative effects were described in eleven individuals from three unrelated families. The probands had variable degrees of short stature, with the laboratory evaluations being suggestive of mild GHI, elevated IgE, and eczema [46]. These data have identified *STAT5B* as a gene that can be associated with ISS.

### 5.1.5. *IGF1* and *IGF1R*

*IGF-1* is essential in intrauterine and postnatal growth, acting through a tyrosine kinase receptor, *IGF-1R*. Patients with LoF homozygous mutations in *IGF1* or *IGF1R* experience prenatal-onset growth impairments, failure to thrive, microcephaly, and developmental delays [76]. While patients with *IGF1* defects have extremely low levels of *IGF-1*, those with *IGF1R* defects show an elevation of this hormone. Both genetic defects cause normal to high concentrations of basal and stimulated GH [76]. The defects in heterozygosity in *IGF1* or *IGF1R* have a much more variable clinical presentation. With the increase in cases with defects in these genes, some patients can be labeled as having ISS. Some cases of heterozygous LoF mutations in *IGF1* were described; these patients had variable degrees of pre- and postnatal growth retardation with serum *IGF-1* at the lower end of the normal

range due to *IGF1* haploinsufficiency [40]. Although most patients fit into the classification of children born small for gestational age (SGA) and have associated microcephaly, some patients may fit into the ISS classification. Heterozygous mutations in *IGF1R* were more frequent than in its ligand but showed similar broad clinical variability levels [77,78]. Some affected patients are born with weight and length within the lower reference limits and develop short stature during childhood, with normal to elevated *IGF-1* values at baseline, which rise above the reference values during GH treatment [50,79]. Thus, heterozygous defects in *IGF1* or *IGF1R* may be present in children classified as having ISS, being more likely in cases when the birth weight or length is at the lower limit of normality.

5.1.6. Ternary Complex Defects (*IGFALS* and *PAPP-A2*)

Most circulating *IGF-1* is found in a ternary complex bound with IGFBP-3 and ALS (acid labile subunit). This complex is important in extending the serum *IGF-1* half-life and decreasing its bioavailability at the tissue level [80]. Homozygous or compound heterozygous defects in the *IGFALS* cause human acid labile subunit (ALS) deficiencies. This condition causes a mild phenotype of GHI, characterized by moderate postnatal growth impairment, pubertal delay, normal or even increased GH, and a marked reduction in the levels of *IGF-1* and IGFBP-3, which remain low after GH treatment [52,80,81] Heterozygous carriers of *IGFALS* gene mutations have a partial ALS deficiency, resulting in less than 1.0 SD of height loss and generally not causing short stature [82]. One study showed that the frequency of rare *IGFALS* variants in children with ISS is 3 times higher than in normal height controls, suggesting that defects in heterozygosity in this gene may contribute to short stature and low *IGF-1* levels in some children with ISS [37].

Also related to the ternary complex, pregnancy-associated plasma protein A2 (*PAPP-A2*) is a serum and tissue protease responsible for the proteolysis of IGFBPs, releasing *IGF-1* from the ternary complex [83]. *PAPP-A2* specifically cleaves IGFBP-3 and -5, so the loss of *PAPP-A2* function leads to increased serum concentrations of IGFBP-3 and IGFBP-5 and total concentrations of *IGF-1*, decreasing the bioactive *IGF-1*. Until now, few families with homozygous LoF mutations in *PAPPA2* have been described [43,84]. In addition to their variable degrees of short stature, some patients had microcephaly, thin long bones, low bone mineral density, and insulin resistance. They also presented with markedly elevated serum concentrations of *IGF-1* and IGFBP-3. As the clinical findings are relatively non-specific, some patients with *PAPPA2* defects may be classified as having ISS if a more detailed hormonal assessment is not performed [83].

*5.2. Genes That Regulate Growth Plate Physiology*

Longitudinal growth depends on the endochondral ossification process in the growth plate. This process involves proliferation, hypertrophy, and the senescence of chondrocytes and cartilage matrix synthesis [85]. The regulation of the entire physiology of the growth plate involves several paracrine and autocrine factors, and defects in genes encoding any of these factors or their respective receptors, inherited in an autosomal dominant manner, can cause a variable phenotype form of isolated short stature with or without mild body disproportion [86]. In the last 20 years, growth-plate-related genes have become protagonists in studies related to ISS [87]. The most consistent and frequent findings were those involving short stature homeobox (*SHOX*) [34], natriuretic peptide receptor 2 (*NPR2*) [39], aggrecan (*ACAN*) [41], Indian hedgehog (*IHH*) [45], natriuretic peptide C (*NPPC*) [44], fibroblast growth factor receptor 3 (*FGFR3*) [42], and collagen-related genes [47]. Each of these genes is responsible for a small proportion of cases of isolated SS (up to 2%), but this proportion may be significantly higher in familial short stature [6,27].

5.2.1. *SHOX*

*SHOX* is located on the pseudoautosomal region of sex chromosomes and is expressed in hypertrophic chondrocytes in the growth plate. *SHOX* is a transcription factor that increases *NPPB* and inhibits the expression of *FGFR3*, stimulating and coordinating the

proliferation and differentiation of chondrocytes. It also has an important role in the synthesis of the growth cartilage matrix through interactions with the *SOX* trio (*SOX9*, *SOX5*, and *SOX6* genes) [49]. Since its first description in 1997, *SHOX* gene haploinsufficiency has become the leading monogenic cause of short stature (1–10% of non-syndromic short stature cases) [34,49].

It is known that defects in this gene comprise an enormous phenotypic variability ranging from isolated short stature with or without mild body disproportion to Leri–Weill dyschondrosteosis (LWD), a type of skeletal dysplasia characterized by mesomelia and Madelung deformity [88]. Despite being recognized, its phenotypic variability still needs to be better understood. Most genetic defects related to the *SHOX* gene (>70%) are deletions involving the whole gene or its regulatory regions. Although the genotype–phenotype correlation is still a matter of debate, deletions in downstream *SHOX* enhancers have been associated with a milder phenotype (variable degrees of short stature, with milder body disproportions and the absence of Madelung deformity) [89]. The growth impairments in *SHOX* haploinsufficiency are variable, but longitudinal follow-up studies suggest the presence of a growth retardation in early life with birth lengths in the lower range of normality, a relatively well-preserved prepubertal growth, an attenuated pubertal growth spurt, and a frequent short adult height. The loss of growth potential observed in these patients suggests an additional effect of estrogens in the presence of *SHOX* defects, which lead to premature fusion of the growth plate. A few studies suggest that recombinant human growth hormone (rhGH) therapy (with or without GnRH analog treatment) can improve the adult height of patients with *SHOX* haploinsufficiency by up to 1 SDS of height [90–93].

### 5.2.2. *NPR2* and *NPPC*

In the last 10 years, the C-type natriuretic peptide (CNP) system and its receptor (NPR-B), encoded by the *NPPC* and *NPR2* genes, respectively, have been identified as important regulators of the endochondral ossification process. The CNP/NPR-B system stimulates the proliferation and differentiation of chondrocytes and the synthesis of the growth plate matrix in an autocrine and paracrine manner. This action is partially explained by the inhibition of the *FGFR3* pathway. Only two families, with short stature and small hand phenotypes, have been described as having heterozygous *NPPC* mutations [44]. In contrast, numerous descriptions of patients with homozygous *NPR2* defects have been made.

Biallelic mutations in *NPR2* lead to acromesomelic dysplasia of the Maroteaux type, a skeletal dysplasia characterized by severe mesomelic short stature and abnormalities in the phalanges and metacarpals [94]. Heterozygous mutations in *NPR2* have been described in cohorts of children with idiopathic short stature (with frequency rates ranging from 1.8 to 13.6%) [82]. These patients have variable degrees of height deficit and progressive short stature with increasing loss of height potential with age [95]. In addition to short stature, these patients may also have mild body disproportion and nonspecific skeletal findings such as short metacarpals [39,96]. Treatment with rhGH promotes short-term height gains, but the data on the final height in treated patients are not yet available [97].

### 5.2.3. *ACAN*

Aggrecan, which is encoded by the *ACAN* gene, is a major proteoglycan component in the extracellular matrix of the growth plate and articular cartilage. Biallelic defects in *ACAN* cause Spondyloepimetaphyseal dysplasia, while alterations in the heterozygous state account for a less severe and more variable phenotype [98]. Since 2014, more than 100 variants have been described in the *ACAN* gene [41,99–102]. Usually, these patients have a severe to mild proportional short stature with advanced bone age in the prepubertal period, which leads to premature growth arrest after the onset of puberty, causing a deficit in adult height. The premature maturation of hypertrophic chondrocytes and the early invasion of growth plates by blood vessels and osteoblasts are proposed mechanisms for advanced bone age, early epiphyseal fusion, and premature growth arrest in patients with

*ACAN* mutations [41]. Despite being a striking feature, it is not found in all patients [103]. Due to its presence in the articular cartilage, the patients may also present with early-onset osteoarthritis, which is more common by the fourth decade of life, mainly affecting the knees, with variable degrees of severity [102]. The limited data about growth patterns show that pubertal spurts may be impaired due to advancing bone age, which lead to a short adult height. Thus, these patients could benefit from puberty blocking and rhGH therapy [99,104].

### 5.2.4. *IHH*

The Indian hedgehog gene (*IHH*) is expressed in prehypertrophic cartilage chondrocytes and coordinates chondrocyte proliferation and differentiation during endochondral bone development [93]. Defects in the *IHH* gene have already been recognized as a cause of skeletal dysplasias in homozygosis (acrocapitofemoral dysplasia) and in heterozygosis (brachydactyly type A1, BDA1) [94]. More recently, heterozygous *IHH* mutations have been reported as a very frequent cause of short stature (frequency of 3.4%), with mild body disproportion and non-specific skeletal abnormalities such as brachymesophalangia-V (BMP-V, shortening of the middle phalanx of the fifth digit of varying degree) [33,95]. Despite being frequent in the general population (12.1%), BMP-V is a very common finding (64.3%) in hand radiographs of individuals with heterozygous mutations in *IHH* [33]. While the *IHH* mutations responsible for BDA1 are always missense and in the same domain of the protein, the mutations causing ISS are distributed throughout the gene and can also be LoF, suggesting a mechanism of haploinsufficiency. A short-term good response to rhGH has been described in these patients [33].

### 5.2.5. *FGFR3*

The fibroblast growth factor receptor-3 (*FGFR3*) pathway acts as a negative regulator of growth plate chondrogenesis. Heterozygous gain-of-function mutations in *FGFR3* are recognized as causing achondroplasia and hypochondroplasia. Achondroplasia is clinically characterized by severe disproportionate short stature with rhizomelia [105]. On the other hand, hypochondroplasia is generally less severe and has a wider phenotypic variability [106]. Although body disproportion is expected in patients with hypochondroplasia, some children may not be diagnosed in the first evaluation. These children may not yet have radiographic findings suggestive of a skeletal defect or may have a slight body disproportion, which can only be recognized through the sitting height/total height ratio. As such, they are often classified as having idiopathic short stature, demonstrating the variable spectrum of the phenotype [107]. Corroborating this hypothesis, a recent study identified an activating mutation of *FGFR3* causing autosomal proportional familial short stature without other specific findings [42].

### 5.2.6. *COL2A1* and Other Collagen Genes

Collagens are the main structural components of the extracellular matrix, with type II collagen being the main collagen synthesized by chondrocytes. This type of collagen is found in the vitreous humor, inner ear, nucleus pulposus, and mainly in the growth cartilage, and is essential for the endochondral ossification process. Mutations in the *COL2A1* gene interfere with the grouping of type II collagen molecules, causing varying degrees of skeletal dysplasia and some types of eye disorders. More than 16 syndromic conditions have been associated with collagen gene defects [108]. Some of these may present with short stature and unspecific skeletal alterations. However, due to the great variability and overlapping phenotypes, mainly in *COL2A1*, *COL11A1*, and *COL11A2*-related disorders, some mutations in collagen genes have already been reported in ISS cohorts [12,31,109–111]. In a recent study, Plachy et al. found 9 collagen genes variants (*COL2A1*, *COL11A1*) in 17 patients (11.5%) in a cohort of familial short stature patients that were treated with growth hormone therapy, reinforcing the role of these genes in genetic causes of short stature usually classified as idiopathic [47].

### 5.3. Genes Related to RAS-MAPK Pathway

Increased signal transduction through the mitogen-activated protein kinase (RAS-MAPK) cascade is a common molecular mechanism that characterizes a heterogeneous group of syndromes classified as RASopathies. Noonan syndrome is the main representative of this group and has short stature as one of its clinical criteria. However, there is a wide spectrum of genetic variability among individuals, and their phenotype–genotype correlations are not fully established, even within the same family. Another challenge in the diagnosis is that even in the presence of typical dysmorphisms, they can undergo changes according to age group, which often makes the clinical diagnosis even more difficult; these children can be classified as having idiopathic short stature due to their attenuated phenotype. Pathogenic variants in genes related to the RAS-MAPK pathway have already been identified in cohorts of patients with idiopathic short stature, with the diagnosis often being only possible using a genetic approach.

### 5.3.1. PTPN11

The protein–tyrosine phosphatase non-receptor type 11 gene (*PTPN11*) encodes the non-receptor Src-homology 2 (SH2) domain-containing protein tyrosine phosphatase 2 (SHP2). SHP2 participates in multiple intracellular signaling pathways, including the Ras/MAPK cascade. Heterozygous mutations in *PTPN11* cause Noonan syndrome (NS), characterized by reduced postnatal growth, congenital heart disease, and facial dysmorphisms [112]. Usually, children can also present with low serum *IGF-1* levels, which reflects some degree of GH insensitivity [113]. Although most children harboring *PTPN11* variants have some clinical features of Noonan syndrome, there is a wide variability in clinical presentation, even in patients with the same genetic variant [100]. In addition, some mutations have a less intense effect (hypomorphic) and a more discrete phenotype [114]. Based on this, *PTPN11* variants have been increasingly described in studies evaluating multiple genes in ISS in children without other clinical criteria of Noonan syndrome [6,11,12,54]. This can be partially explained by the important changes in craniofacial features with age, which sometimes can make a clinical diagnosis of NS challenging. Children with *PTPN11* variants in the milder end of the NS phenotypic spectrum can also be classified as having ISS. The identification of a pathogenic variant in *PTPN11* in a child with isolated short stature has evident impacts on the follow-up and treatment.

### 5.3.2. NF1

Neurofibromin 1 (*NF1*) is a gene related to the RAS-MAPK pathway that encodes neurofibromin, a RAS-specific guanosine triphosphatase (GTPase)-activating protein that converts active GTP-bound RAS to inactive GDP-bound RAS. The inactivation of neurofibromin leads to the hyperactivation of RAS and contributes through other mediators to cell and growth proliferation related to the formation of tumors. The classic presentation of patients who carry variants in the *NF1* gene includes café au lait spots, cutaneous or plexiform neurofibromas, and Lisch nodules in the iris. Intellectual disabilities, vasculopathies, optic gliomas, and increased risk of nervous system tumors, and short stature may be present, which can often suggest a Noonan-like phenotype [115]. Due to having the same phenotypic variability and the fact that the clinical features can change during childhood, *NF1* variants have been found in cohorts of children with ISS, making it a gene to be enrolled in the genetic investigation of patients with isolated phenotypes [54]. Additionally, these data highlight the importance of a genetic diagnosis, as it has a great impact on the clinical follow-up as well as on the decision to start growth hormone therapy in these patients.

## 6. Conclusions and What to Expect from Genomic Investigations

In the last 20 years, the investigations of children with short stature have been changing considerably. The genetic approach has gained importance and emerged as a potential method to establish the etiology of growth impairment. Since the 1950s, when the kary-

otype technique was developed and explained chromosomal abnormalities such as Down's syndrome and Turner's syndrome, a lot has changed. The chromosome microarray analysis (CMA) technique became frequently used due to its greater sensitivity in detecting submicroscopic copy number variants (CNVs, deletions, and duplications). A CMA can reveal a pathogenic CNV in approximately 10% of growth disorders, mainly in children with syndromic short stature of an unknown cause [116]. The identification of point variants (single-nucleotide variants and small insertions and deletions) has also undergone great developments, migrating rapidly from Sanger sequencing to next-generation sequencing (NGS). This new technology allows the sequencing of multiple regions and genes simultaneously and can include an analysis of CNVs, an in-target gene panel, and whole-exome sequencing (WES) or whole-genome sequencing (WGS).

The use of NGS-based techniques improves the diagnostic yield and better explains the genetic basis of the isolated short stature phenotype for patients initially classified as having ISS. Although the current guidelines suggest that the genetic test should preferably be conducted in syndromic conditions or when it is possible to determine a candidate gene [22,117], this approach is being challenged. Children classified as having ISS infrequently show signs that reliably allow a candidate gene investigation, and for this reason the multigene approach is a logical strategy to assess them. Additionally, it has been increasingly shown that a significant number of children with ISS have a monogenic form of short stature (Table 2). The average diagnostic rate of the multigene approach in ISS is around 12.5% (ranging from 8.7 to 19.5%) [12]. Finally, the costs of genetic tests and their analysis have demonstrated substantial and rapid improvements. The perspective is that in the near future, according to Moore's law, the price of human genome sequencing may drop quickly and considerably [118]. Consequently, a multigene approach (e.g., a targeted gene panel or WES) in children with ISS will soon become a good option, since it might save time and costs when used with other exams and provide an answer for a significant number of families about the cause of short stature, allowing the rapid identification of other affected family members.

Additionally, knowing the genetic basis of these patients' short stature can allow precise genetic counseling and clinical follow-up. This is particularly relevant to children harboring variants in the RAS-MAPK pathway. Moreover, the use of rhGH therapy in ISS is controversial, and at some point the genetic studies can support the use of this therapy. It is accepted that children with *SHOX* haploinsufficiency can improve their adult height with rhGH therapy [91,93]. Some preliminary data have already associated with a good short-term response to rhGH treatment in patients with variants in the *IHH* [45], *NPR2* [97], and *ACAN* [119] genes. The knowledge of the genetic basis of ISS has the potential to clarify the growth pattern and to trigger the development of specific treatment protocols, including the indication and the use of rhGH, natriuretic peptide analogs, or puberty modulators with the added benefit of permitting drug development through targeted therapy.

However, a substantial number of children with ISS remain undiagnosed, even after performing a WES analysis. Initially, we must remember that not every short child will be a short adult. Observational studies of children with ISS show that the majority show spontaneous catch-up growth [120]. These children are not expected to have genetic factors (mono- or polygenic) that cause short stature. Secondly, new genes associated with ISS need to be revealed. Furthermore, as height is a polygenic trait, it is expected that using a polygenic risk score (PRS) in combination with rare variants may increase the degree of knowledge about height determination, allowing the application of the concept of precision medicine in patients with ISS [26]. What we expect from the genomic investigation is that these diagnostic tools will be routinely offered and recommended in assessing children with ISS, so that we can close those gaps that remain open to so many families who seek our help.

**Table 2.** A review of multigene approach studies on "idiopathic short stature".

| Reference | Total Cohort | ISS Cohort | Methodology | Diagnostic Yield (Total Cohort) | Diagnostic Yield (ISS Cohort) | Discussion |
|---|---|---|---|---|---|---|
| Wang et al. (2013) [6] | 192 | 14 | Target gene panel with 1077 genes | 4/192 (2%) | 3/14 (21.4%) | Included syndromic children |
| Guo et al. (2014) [7] | 14 | 3 | Exome | 5/14 (35.7%) | 0 | Only genes related to syndromic short stature |
| Hattori et al. (2017) [8] | 86 | 86 | Target gene panel with 10 genes | 18 (20.9%) | 18 (20.9%) | Excluded SGA and syndromic children |
| Hauer et al. (2018) [9] | 200 | 13 | Exome | 38/200 (19%) | 11/13 (84.6%) | 134 children initially classified as having ISS but included children born SGA and with syndromic features |
| Freire et al. (2019) [10] | 179 | 55 | Target gene panel/exome | 8/55 (14.5%) | 8/55 (14.5%) | Included only children with isolated short stature born SGA |
| Perchard et al. (2020) [11] | 263 | 18 | Target gene panel | 27/263 (10%) | 5/18 (27.8%) | Included children with GH deficiency and dysmorphic features |
| Fan et al. (2021) [13] | 561 | 257 | Exome | 135/561 (24%) | 11.3% (29/257) | Included genes associated with hypopituitarism, skeletal dysplasia, and chronic and metabolic disorders with the positive results in the ISS group. |
| Sentchordi-Montané et al. (2021) [12] | 108 | 108 | Target gene panel | 21/108 (19.4%) | 12/108 (11.1%) | Skeletal dysplasia NGS panel and *SHOX*-related genes were excluded. Included children born SGA. |
| Andrade et al. (2022) [121] | 102 | 102 | Target gene panel | 17/102 (16.7%) | 17/102 (16.7%) | Only children classified as having ISS |

**Author Contributions:** Conceptualization, A.A.L.J.; methodology and selection of manuscripts to be included in the review, A.A.L.J. and N.L.M.A.; writing—original draft preparation, N.L.M.A. and L.P.C.; writing—review and editing, R.C.R., G.A.V. and A.A.L.J.; supervision, A.A.L.J. and G.A.V. All authors have read and agreed to the published version of the manuscript.

**Funding:** This work was supported by Grant 303294/2020-5 (to A.A.L.J.) from the National Council for Scientific and Technological Development (CNPq) and by Coordination of Superior Level Staff Improvement (CAPES; Finance Code 001 to N.L.M.A., L.P.C. and R.C.R.).

**Institutional Review Board Statement:** Not applicable.

**Informed Consent Statement:** Not applicable.

**Data Availability Statement:** Not applicable.

**Conflicts of Interest:** A.A.L.J. received an independent research grant from BioMarin and consulting fees from Novo Nordisk. The other authors declare that they have no competing financial interest to declare.

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
