# Peer review of "Idiopathic Short Stature: What to Expect from Genomic Investigations"

_endocrines, doi:10.3390/endocrines4010001_

Round 1
Reviewer 1 Report
I read the narrative review of Nathalia L. M. Andrade et al tiled “Idiopathic short stature: what to expect from genomic investigation”
I find it of clinical interest althought it has some limitations
-
I recommend following SANRA (PMID: 30962953) for improve the quality of the review: Specifically, the description of the literature search (item 3) is lacking.
-
The conclusion must be improved and toned down to give a more realistic clinical perspective. The approach to genetic testing in short stature is very complex and a helpful algorithm is lacking. Although the authors are in favour of NGS-based techniques, it is important to differentiate the multigene approach ( test all the genome from a potential individual) from an indiscriminate genetic testing of all the children with ISS. Without an algorithm to guide who to test, the results are likely to give more uncertainty than answers.
-
What will happen with variant of uncertain clinical significance?
-
What can and should we do therapeutically? in the own individual, in the siblings and offspring
Author Response
Reviewer 1
The authors thank the reviewer for the comments and relevant suggestions. Changes are highlighted in yellow in the revised manuscript.
Review: “I recommend following SANRA (PMID: 30962953) for improve the quality of the review: Specifically, the description of the literature search (item 3) is lacking.”
Answer: Thanks for this recommendation. The topic of materials and methods was added with the search strategy and eligibility criteria for selecting articles.
Review: “The conclusion must be improved and toned down to give a more realistic clinical perspective. The approach to genetic testing in short stature is very complex and a helpful algorithm is lacking. Although the authors are in favour of NGS-based techniques, it is important to differentiate the multigene approach ( test all the genome from a potential individual) from an indiscriminate genetic testing of all the children with ISS. Without an algorithm to guide who to test, the results are likely to give more uncertainty than answers”.
Answer: The conclusion was modified in some topics and new references were added.
Review: “What will happen with variant of uncertain clinical significance?”
Answer: We believe that variants of uncertain significance should probably be interpreted with caution and that most will be negative genetic test results and should not be reported. I suggest below an editorial that addresses this issue as we do in our practice.
Morales A, Hershberger RE. Variants of Uncertain Significance: Should We Revisit How They Are Evaluated and Disclosed? Circ Genom Precis Med. 2018 Jun;11(6):e002169. doi: 10.1161/CIRCGEN.118.002169. PMID: 29848615; PMCID: PMC5999032.
Review: “What can and should we do therapeutically? in the own individual, in the siblings and offspring”
Answer: We added a paragraph in the discussion about the therapeutic approach

Reviewer 2 Report
This manuscript is well organized around genetic abnormalities in the broad area of idiopathic short stature.
I am confident that it will be helpful to the reader's understanding of the disease.
One point that should be modified is that the numbers indicating references should be superscripted.
Author Response
Reviewer 2
The authors thank the reviewer for the comments and relevant suggestions. Changes are highlighted in yellow in the revised manuscript.
Review: “ This manuscript is well organized around genetic abnormalities in the broad area of idiopathic short stature.
I am confident that it will be helpful to the reader's understanding of the disease.”
Answer: Thank you for your comment.
Review: “One point that should be modified is that the numbers indicating references should be superscripted”
Answer: Some references were changed and all of them superscripted.

Round 2
Reviewer 1 Report
I accept in present form